# *Spoink*, a LTR retrotransposon, invaded *D. melanogaster* populations in the 1990s

**Riccardo Pianezza**[1,2☯], **Almorò Scarpa**[1,2☯], **Prakash Narayanan**[3], **Sarah Signor**[3]*, **Robert Kofler**[1]*

**1** Institut für Populationsgenetik, Vetmeduni Vienna, Vienna, Austria, **2** Vienna Graduate School of Population Genetics, Vetmeduni Vienna, Vienna, Austria, **3** Biological Sciences, North Dakota State University, Fargo, North Dakota, United States of America

☯ These authors contributed equally to this work.
* sarah.signor@ndsu.edu (SS); rokofler@gmail.com (RK)

## Abstract

During the last few centuries *D. melanogaster* populations were invaded by several transposable elements, the most recent of which was thought to be the *P*-element between 1950 and 1980. Here we describe a novel TE, which we named *Spoink*, that has invaded *D. melanogaster*. It is a 5216nt LTR retrotransposon of the Ty3/gypsy superfamily. Relying on strains sampled at different times during the last century we show that *Spoink* invaded worldwide *D. melanogaster* populations after the *P*-element between 1983 and 1993. This invasion was likely triggered by a horizontal transfer from the *D. willistoni* group, much as the *P*-element. *Spoink* is probably silenced by the piRNA pathway in natural populations and about 1/3 of the examined strains have an insertion into a canonical piRNA cluster such as *42AB*. Given the degree of genetic investigation of *D. melanogaster* it is perhaps surprising that *Spoink* was able to invade unnoticed.

## Author summary

Horizontal transfer of transposable elements (TE) is a major factor driving genome evolution. Yet well documented cases of such horizontal transfer events are rare. Most evidence is indirect, relying on sequence similarity of TEs between species. Based on strains sampled during the last decades we provide direct evidence that the retrotransposon *Spoink* was absent in worldwide *D. melanogaster* populations before 1983 but present in populations after 1993. We suggest that the *Spoink* invasion was triggered by a horizontal transfer from a *Drosophila* species of the *willistoni* group.

## Introduction

Transposable elements (TEs) are short genetic elements that can increase in copy number within the host genome. They are abundant in most organisms and can make up the majority of some genomes, i.e. maize where TEs constitute 83% of the genome [1]. There are two classes of TEs which transpose by different mechanisms—DNA transposons which replicate by

**Data Availability Statement:** The consensus sequence of Spoink as well as the sequences of the six PCR amplicons are available at https://github.com/rpianezza/Dmel-Spoink/tree/main/

releasedseqs. The tool LTRtoTE is available on GitHub (https://github.com/Almo96/LTRtoTE). The analysis performed in this work have been documented with RMarkdown and have been made publicly available, together with the resulting figures, at GitHub (https://github.com/rpianezza/Dmel-Spoink; see *.md files).

**Funding:** This work was supported by the National Science Foundation Established Program to Stimulate Competitive Research grants NSF-EPSCoR-1826834 and NSF-EPSCoR-2032756 to SS, and by the Austrian Science Fund (FWF) grants P35093 and P34965 to RK. The funders had no role in study design, data collection and analysis, decision to publish, or preparation of the manuscript.

**Competing interests:** The authors have declared that no competing interests exist.

directly moving to a new genomic location in a 'cut and paste' method, and retrotransposons which replicate through an RNA intermediate in a 'copy and paste' method [2–4]. From humans to flies, more genetic variation (in bp) is due to repetitive sequences such as transposable elements than all single nucleotide variants combined [5]. Although some TEs, such as *R1* and *R2* elements, may benefit hosts [6, 7] most TE insertions are thought to be deleterious [8, 9]. Host genomes have therefore evolved an elaborate system of suppression frequently involving small RNAs [10]. Suppression of TEs in *Drosophila* relies upon small RNAs termed piRNA, which are cognate to TE sequences [11–13]. These small RNAs bind to PIWI clade proteins and mediate the degradation of TE transcripts and the formation of heterochromatin silencing the TE [11, 14–19]. However, while host defenses quickly adapt to new transposon invasions, TEs can escape silencing through horizontal transfer to new, defenseless, genomes [20–23]. This horizontal transfer allows TEs to colonize the genomes of novel species [20, 23–26]. The first well-documented instance of horizontal transfer of a TE was the *P*-element, which spread from *D. willistoni* to *D. melanogaster* [27]. Following this horizontal transfer the *P*-element invaded natural *D. melanogaster* populations between 1950 and 1980 [28, 29]. It was further realized that the *I*-element, *Hobo* and *Tirant* spread in *D. melanogaster* populations earlier than the *P*-element, between 1930 and 1960 [29–31]. The genomes from historical *D. melanogaster* specimens collected about two hundred years ago, recently revealed that *Opus*, *Blood*, and *412* spread in *D. melanogaster* populations between 1850 and 1933 [21]. In total, it was suggested that seven TEs invaded *D. melanogaster* populations during the last two hundred years where one invasion (the *P*-element) was triggered by horizontal transfer from a species of the *willistoni* group and six invasions by horizontal transfer from the *simulans* complex [21, 27, 31–34].

It was, however, widely assumed until now that the *P*-element invasion, which occurred between 1950–1980, was the last and most recent TE invasion in *D. melanogaster* [21, 29, 31, 35, 36]. Here we report the discovery of *Spoink*, a novel TE which invaded worldwide *D. melanogaster* populations between 1983 and 1993, i.e. after the invasion of the *P*-element. *Spoink* is a LTR retrotransposon of the Ty3/gypsy group. We suggest that the *Spoink* invasion in *D. melanogaster* was triggered by horizontal transfer from a species of the *willistoni* group, similarly to the *P*-element invasion in *D. melanogaster*. In a model species as heavily investigated as *D. melanogaster* it is perhaps surprising that *Spoink* was able to invade undetected.

## Materials and methods

### Discovery of the recent *Spoink* invasion

We identified TE insertions in different long-read assemblies using RepeatMasker [37] and the TE library from [5]. When comparing the TE composition between strains collected in the 1950's and 1960's [38, 39] and more recently collected strains (≥ 2003 [40] we noticed an element labeled 'gypsy-7_DEl' which was only present in short degraded copies in the older genomes but was present in full length copies in the more recent genomes (S1 Table).

### Structure and classification of *Spoink*

To generate a consensus sequence of *Spoink* we extracted the sequence of full-length matches of '*gypsy-7_DEl*' plus some flanking sequences from long-read assemblies [Ten-15, RAL91, RAL176, RAL732, RAL737, Sto-22; [40]] and made a consensus sequence by performing multiple sequence alignment (MSA) with MUSCLE (v3.8.1551) [41] and then choosing the most abundant nucleotide in each position of the MSA with a custom Python script (MSA2consensus).

The consensus sequence of the LTR was used to identify the TSD with our new tool LTRtoTE (https://github.com/Almo96/LTRtoTE). We used LTRdigest to identify the PBS of *Spoink* [42].

We picked several sequences from each of the known LTR superfamily/groups using the consensus sequences of known TEs [2, 43] (v9.44). We performed a blastx search against the NCBI database to identify the RT domain in the consensus sequences of the TE [44]. We then performed a multiple sequence alignment of the amino-acid sequences of the RT domain using MUSCLE (v3.8.1551) [41]. We obtained the xml file using BEAUti2 [45] (v2.7.5) and generated the trees with BEAST (v2.7.5) [45]. The maximum credibility tree was built using TreeAnnotator (v2.7.5) [45] and visualized with FigTree (v1.4.4, http://tree.bio.ed.ac.uk/software/figtree/).

## Distribution of *Spoink* insertions

Genes were annotated in each of the 31 genomes from [40] using the annotation of the reference genome of *D. melanogaster* (6.49; Flybase) and liftoff 1.6.3 [46, 47]. The 1kb regions upstream of each gene were classified as putative promotors. The location of canonical *D. melanogaster* piRNA clusters was determined using CUSCO, which lifts over the flanks of known clusters in a reference genome to locate the homologous region in a novel genome [48]. The location of *Spoink* insertions within genes or clusters was determined with bedtools intersect [49]. To determine if genic insertions were shared or independent, the sequence of the insertion was extracted from each genome along with an extra 1 kb of flanking sequence on each end. Insertions purportedly in the same gene were then aligned, and if the flanks aligned they were considered shared insertions. To determine if cluster insertions were shared the flanking TE regions were aligned using Manna, which aligns TE annotations rather than sequences, to determine if there was any shared synteny in the surrounding TEs [50].

## Abundance of *Spoink* insertions in different *D. melanogaster* strains

We investigated the abundance of *Spoink* in multiple publicly available short-read data sets [31, 40, 51–53]. These data include genomic DNA from 183 *D. melanogaster* strains sampled at different geographic locations during the last centuries. For an overview of all analysed short-read data see S5 Table. We mapped the short reads to a database consisting of the consensus sequences of TEs [43] (v9.44), the sequence of *Spoink* and three single copy genes (*rhi*, *tj*, *RpL32*) with bwa bwasw (version 0.7.17-r1188) [54]. We used DeviaTE (v0.3.8) [55] to estimate the abundance of *Spoink*. DeviaTE estimates the copy number of a TE (e.g. *Spoink*) by normalizing the coverage of the TE by the coverage of the single copy genes. We also used DeviaTE to visualize the abundance and diversity of *Spoink* as well as to compute the frequency of SNPs in *Spoink* (see below).

To identify *Spoink* insertions in 49 long-read assemblies of *D. melanogaster* strains collected during the last 100 years we used RepeatMasker [37] (open-4.0.7; -no-is -s -nolow). For an overview of all analysed assemblies see S6 Table [39, 40, 48, 56]. For estimating the abundance of *Spoink* in the long-read assemblies we solely considered canonical *Spoink* insertions ($> 80\%$ of length, $< 5\%$ sequence divergence).

## Population frequency of *Spoink* insertions

For every putative *Spoink* insertion (including degraded ones) in the eight long-read assemblies of individuals from Raleigh [40], we extracted the sequence of the insertion plus 1 kb of flanking sequence with bedtools [49]. The sequence of the *Spoink* insertion was removed with seqkit [57] and the flanking sequences were mapped to the *AKA017* genome (i.e. the common

coordinate system) with minimap2 allowing for spliced mappings [40, 57, 58]. The mapping location of each read was extracted and if they overlapped between strains they were considered putative shared sites. Regions with overlapping reads were visually inspected in IGV (v2.4.14) and if the mapping location was shared they were considered shared insertions sites [59, 60].

## PCR

To validate whether *Spoink* is absent in old *D. melanogaster* strains but present in recent strains we used PCR. We designed two primers pairs for *Spoink* and one for *vasa* as a control. We extracted DNA from different strains of *D. melanogaster* (*Lausanne-S, Hikone-R, iso-1, RAL59, RAL176, RAL737*) using a high salt extraction protocol [61]. We designed two primers pairs for *Spoink* (P1,P2) and one for the gene *vasa* (P1 FWD TCAGAAGTGGGATCGGGCTCGG, P1 REV CAGTAGAGCACCATGCCGACGC, P2 FWD ATGGACCGTAATGGCAGCAGCG, P2 REV ACACTCCGCGCCAGAGTCAAAC, Vasa FWD AACGAGGCGAGGAAGTTTGC, Vasa REV GCGATCACTACATGGCAGCC). We used the following PCR conditions: 1 cycle of 95°C for 3 minutes; 33 cycles of 95°C for 30 seconds, 58°C for 30 seconds and 72°C for 20 seconds; 1 cycle of 72°C for 6 minutes.

## Small RNAs

To identify piRNAs complementary to *Spoink* we analysed the small-RNA data from 10 GDL strains [62]. The adaptor sequence GAATTCTCGGGTGCCAAGG was removed using cutadapt (v4.4 [63]). We filtered for reads having a length between 18 and 36nt and aligned the reads to a database consisting of *D. melanogaster* miRNAs, mRNAs, rRNAs, snRNAs, snoRNAs, tRNAs [64], and TE sequences [43] with novoalign (v3.09.04). We used previously developed Python scripts [65] to compute ping-pong signatures and to visualize the piRNA abundance along the sequence of *Spoink*.

## UMAP

We used the frequencies of SNPs in the sequence of *Spoink* to compute the UMAP. This frequencies reflect the *Spoink* composition in a given sample. For example if a specimen has 20 *Spoink* insertions and a biallelic SNP with a frequency of 0.8 at a given site in *Spoink* than about 16 *Spoink* insertions will have the SNP and 4 will not have it. The frequency of the *Spoink* SNPs was estimated with DeviaTE [55]. Solely bi-allelic SNPs were used and SNPs only found in few samples were removed (≤3 samples). UMAPs were created in R (umap package; v0.2.10.0 [66]).

## Origin of horizontal transfer

To identify the origin of the horizontal transfer of *Spoink* we used RepeatMasker [37] (open-4.0.7; -no-is -s -nolow) to identify sequences with similarity to *Spoink* in the long-read assemblies of 101 drosophilid species and in 99 different insect species [67, 68] (S8 Table). We included the long-read assembly of the *D. melanogaster* strain *RAL737* and of the *D. simulans* strain SZ129 in the analysis [23, 40]. We used a Python script to identify in each assembly the best hit with *Spoink* (i.e. the highest alignment score) and than estimated the similarity between this best hit and *Spoink*. The similarity was computed as $s = rms_{best}/rms_{max}$, where $rms_{best}$ is the highest RepeatMasker score (rms) in a given assembly and $rms_{max}$ the highest score in any of the analysed assemblies. A $s = 0$ indicates no similarity to the consensus sequence of *Spoink* whereas $s = 1$ represent the highest possible similarity. To generate a

phylogenetic tree we identified *Spoink* insertions in the assemblies of the 101 drosophilid species and *RAL737* using RepeatMasker. We extracted the sequences of full-length insertions (> 80% of the length) from species having at least one full-length insertion using bedtools [49] (v2.30.0). A multiple sequence alignment of the *Spoink* insertions was generated with MUS-CLE (v3.8.1551) [41] and a tree was generated with BEAST (v2.7.5) [45].

## Results

Previous work showed that at least seven TE families invaded *D. melanogaster* populations during the last two hundred years [21, 29, 31]. To explore whether additional, hitherto poorly characterised TEs could have invaded *D. melanogaster*, we investigated long-read assemblies of recently collected *D. melanogaster* strains [40] using a newly assembled repeat library [5]. Interestingly we found differences in the abundance of "gypsy_7_DEl" between the reference strain *Iso-1* and more recently collected *D. melanogaster* strains (S1 Table). To better characterize this TE, we generated a consensus sequence based on the novel insertions and checked if this consensus sequence matches any of the repeats described in repeat libraries generated for *D. melanogaster* and related species [5, 40, 43, 69, 70]. A fragmented copy of this TE, with just one of the two LTRs being present, was reported by [40] (0.13% divergence; "con41_-UnFmcl001_RLX-incomp"; S2 Table). The next best hits were *gypsy7 Del*, *gypsy2 DSim*, *micropia* and *Invader6* (18–30% divergence; S2 Table). Given this high sequence divergence from previously described TE families and the fact that this novel TE belongs to an entirely different superfamily/group than *gypsy7* (see below), we decided to give this TE a new name. We call this novel TE "*Spoink*" inspired by a Pokémon that needs to continue jumping to stay alive.

*Spoink* is an LTR retrotransposon with a length of 5216 bp and LTRs of 349 bp (Fig 1A; for coordinates of the analysed insertions see S3 Table). At positions 4639–4700 *Spoink* contains a poly-A tract, which length may differ by a few bases between insertions. *Spoink* encodes a 695 aa putative *gag-pol* polyprotein. Ordered from the N- to the C-terminus, the conserved domains of the polyprotein are: reverse transcriptase of LTR (e-value = $2.2e-59$; CDD v3.20 [71]), RNase HI of Ty3/gypsy elements (e-value = $1.65e-48$;) and integrase zinc binding domain (e-value = $4.81e-16$). *Spoink* lacks an *env*. The order of these domains, with the integrase downstream of the reverse transcriptase, is typical for Ty3/gypsy transposons [72].

A phylogeny based on the reverse transcriptase domain of different TE families suggests that *Spoink* is a member of the gypsy/mdg3 superfamily/group of LTR retrotransposons (Fig 1B; [2]). As expected for members of the Ty3/gypsy superfamily, *Spoink* generates a target site duplication of 4 bp and it has an insertion motif enriched for ATAT (Fig 1A; [2, 73]). A *gag-pol* polyprotein as encoded by *Spoink* was observed for some Ty3/gypsy transposons [74, 75] but not for others [72]. However, *Spoink* differs from what is expected for the Ty3/gypsy superfamily in two ways. First, the predicted primer binding site of *Spoink* directly follows the LTR, whereas typically for Ty3/gypsy there is a shift of 5–8nt (Fig 1A; [2]). Second, the LTR motif is TG...TA which is different from the TG...CA motif usually reported for *gypsy* TEs [2].

Finally we investigated the genomic distribution of *Spoink* insertions in long-read assemblies of *D. melanogaster* strains collected ≥ 2003 [40]. In total, these assemblies contains 481 full-length (> 80% length with at least one LTR) insertions of *Spoink* (on the average 16 per genome). Unlike the *P*-element which has a strong insertion bias into promoters, *Spoink* insertions are mostly found in introns and intergenic regions (S1 Fig). 54% of the *Spoink* insertions are in 201 different genes. Interestingly we found 7 independent *Spoink* insertions in *Myo83F*.

To summarize we characterized a novel LTR-retrotransposon of the Ty3/gypsy superfamily in the genome of *D. melanogaster* that we call *Spoink*.

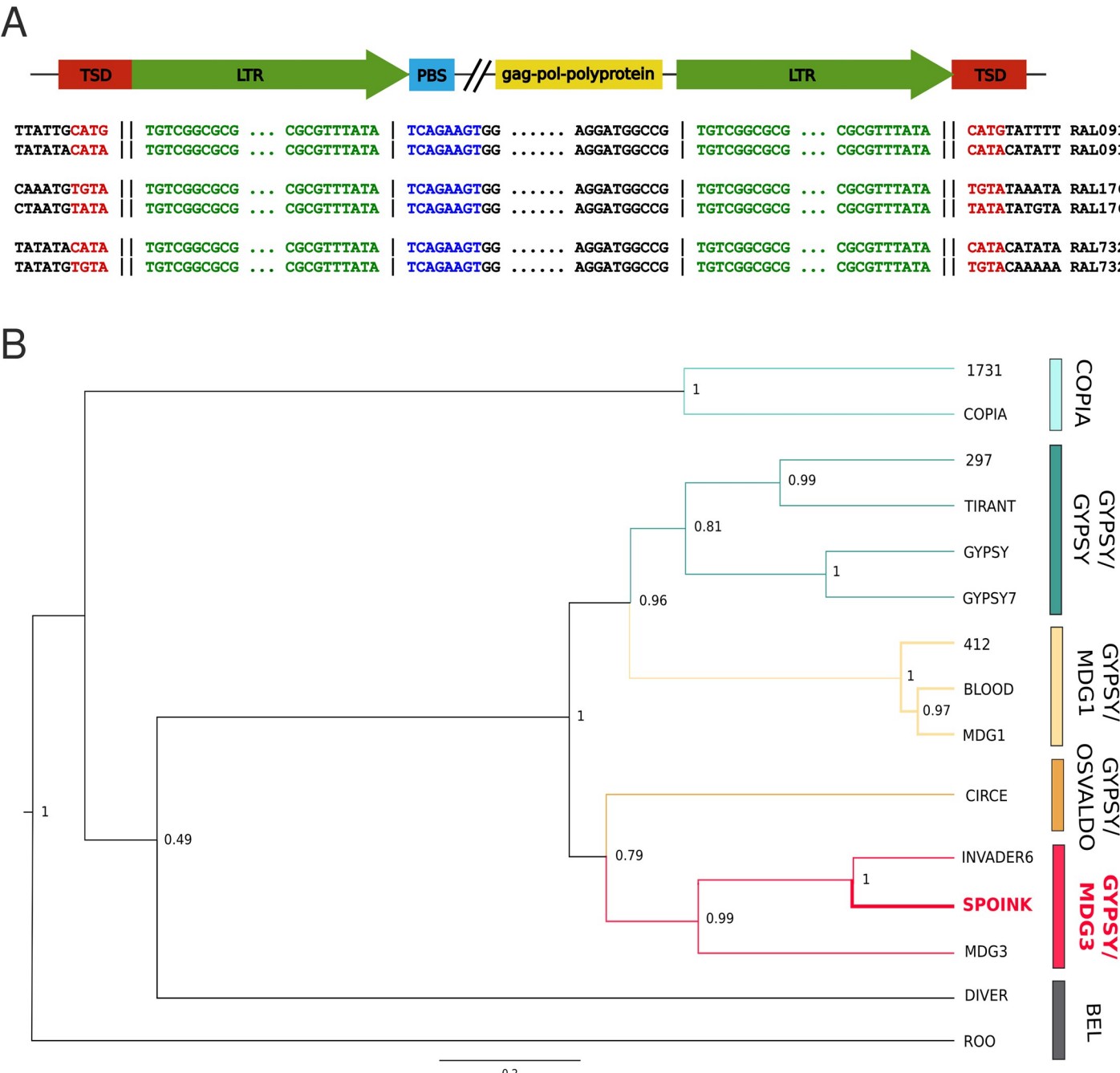

**Fig 1. *Spoink* is a novel TE of the Ty3/gypsy superfamily.** A) Overview of the composition of *Spoink*. Features are shown in color and the alignments show the sequences around the main features of *Spoink* for two insertions in each of three different long-read assemblies of *D. melanogaster*. B) Phylogenetic tree based on the reverse-transcriptase domain of *pol* for *Spoink* and several other LTR retrotransposons. Multiple families have been picked for each of the main superfamilies/groups of LTR transposons [2]. Our data suggest that *Spoink* is a member of the gypsy/mdg3 group.

## *Spoink* recently invaded worldwide *D. melanogaster* populations

To substantiate our hypothesis that *Spoink* recently invaded *D. melanogaster* we used three independent approaches: Illumina short read data, long-read assemblies, and PCR/Sanger sequencing. First we aligned short reads from a strain collected in 1958 (*Hikone-R*) and a strain

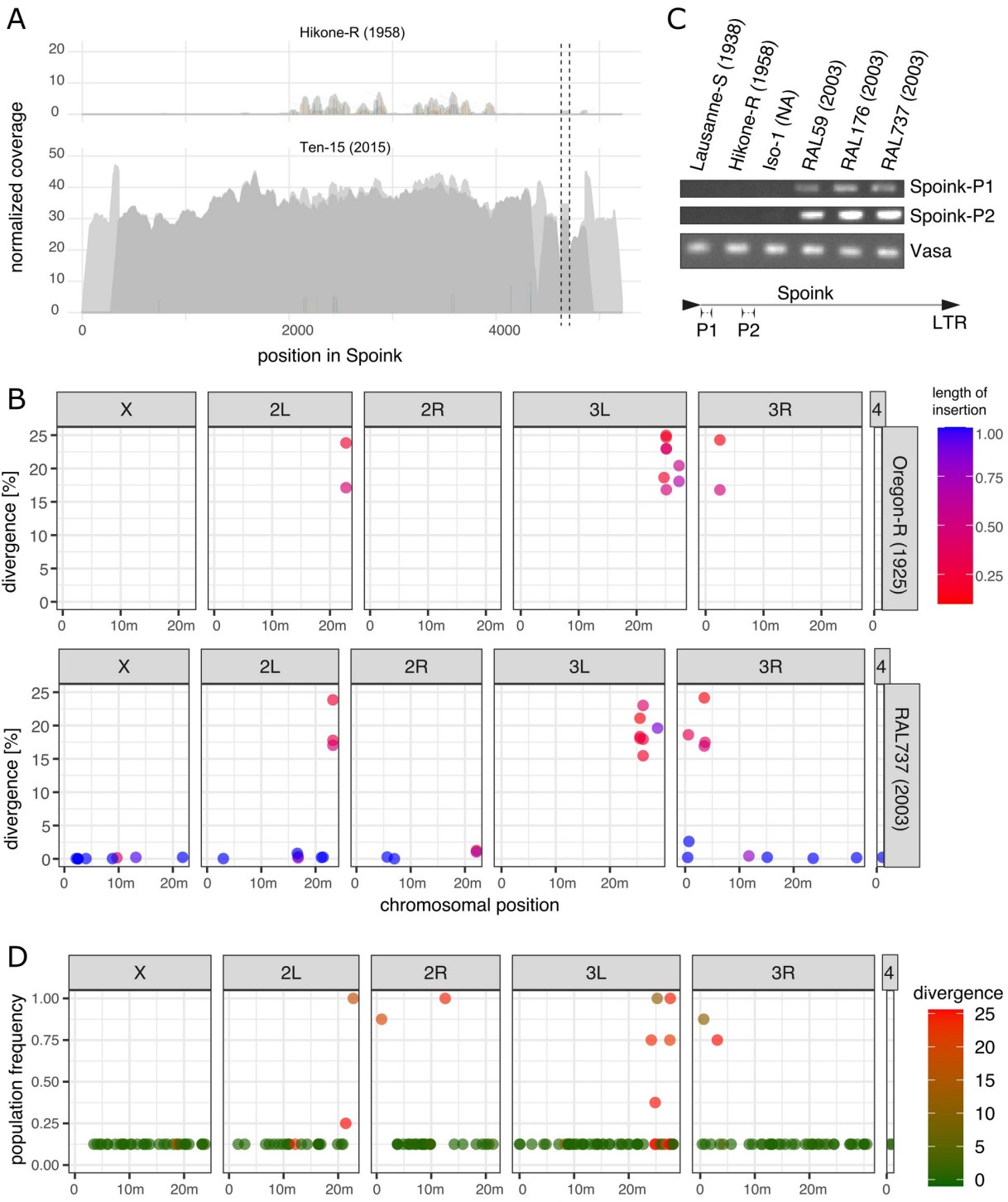

**Fig 2. *Spoink* invaded *D. melanogaster*.** A) DeviaTE plots of *Spoink* for a strain collected in 1954 (*Hikone-R*) and a strain collected in 2015 (*Ten-15*). Short reads were aligned to the consensus sequence of *Spoink* and the coverage was normalized to the coverage of single-copy genes. The coverage based on uniquely mapped reads is shown in dark grey and light grey is used for ambiguously mapped reads. Single-nucleotide polymorphisms (SNPs) and small internal deletions (indels) are shown as colored lines. The coverage was manually curbed at the poly-A track (between dashed lines). B) Insertions with a similarity to the consensus sequence of *Spoink* in the long-read assemblies of *Oregon-R* (collected around 1925) and the more recently collected strain *RAL737* (2003). C) PCR results for two *Spoink* primer pairs (for location of primers see sketch at bottom) and one primer pair for the gene *vasa*. *Spoink* is absent in old strains (*Lausanne-S*, *Hikone-R* and *Iso-1*) and present in more recently collected strains (*RAL59*, *RAL176*, *RAL737*). D) Population frequency of *Spoink* insertions in long-read assemblies of strains collected

in 2003 from Raleigh [40]. Note that highly diverged insertions are largely segregating at a high frequency while canonical *Spoink* insertions mostly segregate at a low frequency.

collected in 2015 (*Ten-15*) [31, 40] to the consensus sequence of *Spoink* using DeviaTE [55]. DeviaTE estimates the abundance of *Spoink* insertions by normalizing the coverage of *Spoink* to the coverage of a sample of single-copy genes. Furthermore, DeviaTE is useful for generating an intuitive visualization of the abundance and composition (i.e. SNPs, indels, truncations) of *Spoink* in samples. We found that only a few degraded reads aligned to *Spoink* in the 1950's strain (*Hikone-R*) whereas many reads covered the sequence of *Spoink* in the more recently collected strain *Ten-15* (Fig 2A). There were also very few SNPs or indels in the recently collected strain suggesting that most insertions have a very similar sequence (Fig 2A). This observation holds true when multiple old and young *D. melanogaster* strains are analysed (S2 Fig).

Next we investigated the abundance of *Spoink* in long-read assemblies of a strain collected in 1925 (*Oregon-R*) and a strain collected in 2003 (*RAL737*). We found solely highly diverged and fragmented copies of sequences with similarity to *Spoink* in *Oregon-R* (Fig 2B). These degraded fragments were mostly found near the centromeres of *Oregon-R*. Investigating the identity of these degraded fragments of *Spoink* in more detail we found that they largely match with short and highly diverged fragments of *Invader6*, *micropia* and the *Max-element* (S4 Table). In addition to these degraded fragments, the more recently collected strain *RAL737* also carries a large number of full-length insertions with a high similarity to the consensus sequence of *Spoink* (henceforth canonical *Spoink* insertions; Fig 2B). The canonical *Spoink* insertions are distributed all over the chromosomes of *RAL737* (Fig 2B). This observation is again consistent when several long-read assemblies of old and young *D. melanogaster* strains are analysed (S3 Fig).

Finally we used PCR to test whether *Spoink* recently spread in *D. melanogaster*. We designed two PCR primer pairs for *Spoink* and, as a control, one primer pair for *vasa* (Fig 2C; bottom panel). The *Spoink* primers amplified a clear band in three strains collected 2003 in Raleigh but no band was found in earlier collected strains, including the reference strain of *D. melanogaster*, *Iso-1* (Fig 2C). We sequenced the fragments amplified by the *Spoink* primers using Sanger sequencing and found that the sequence of the six amplicons matches with the consensus sequence of *Spoink* (S4 Fig).

Finally we investigated the population frequency of canonical and degraded *Spoink* insertions. Using the long-read assemblies of eight strains collected in 2003 in Raleigh we computed the population frequency of different *Spoink* insertions. We found that canonical *Spoink* insertions ($< 5\%$ divergence) are largely segregating at a low population frequency, as expected for recently active TEs (Fig 2D). While several degraded fragments that were annotated as *Spoink* are private, there were many at a higher population frequency as expected for older sequences (Fig 2D).

In summary our data suggest that *Spoink* recently spread in *D. melanogaster* and that degraded fragments with some similarity to *Spoink* are present in heterochromatic regions of the centromeres of all investigated *D. melanogaster* strains. These degraded fragments may be the remnants of more ancient invasions of TEs sharing some sequence similarity with *Spoink*.

## Timing the *Spoink* invasion

Next we sought to provide a more accurate estimate of the time when *Spoink* spread in *D. melanogaster*. First we generated a rough timeline of the *Spoink* invasion using *D. melanogaster* strains sampled during the last two hundred years. We estimated the abundance of *Spoink* in

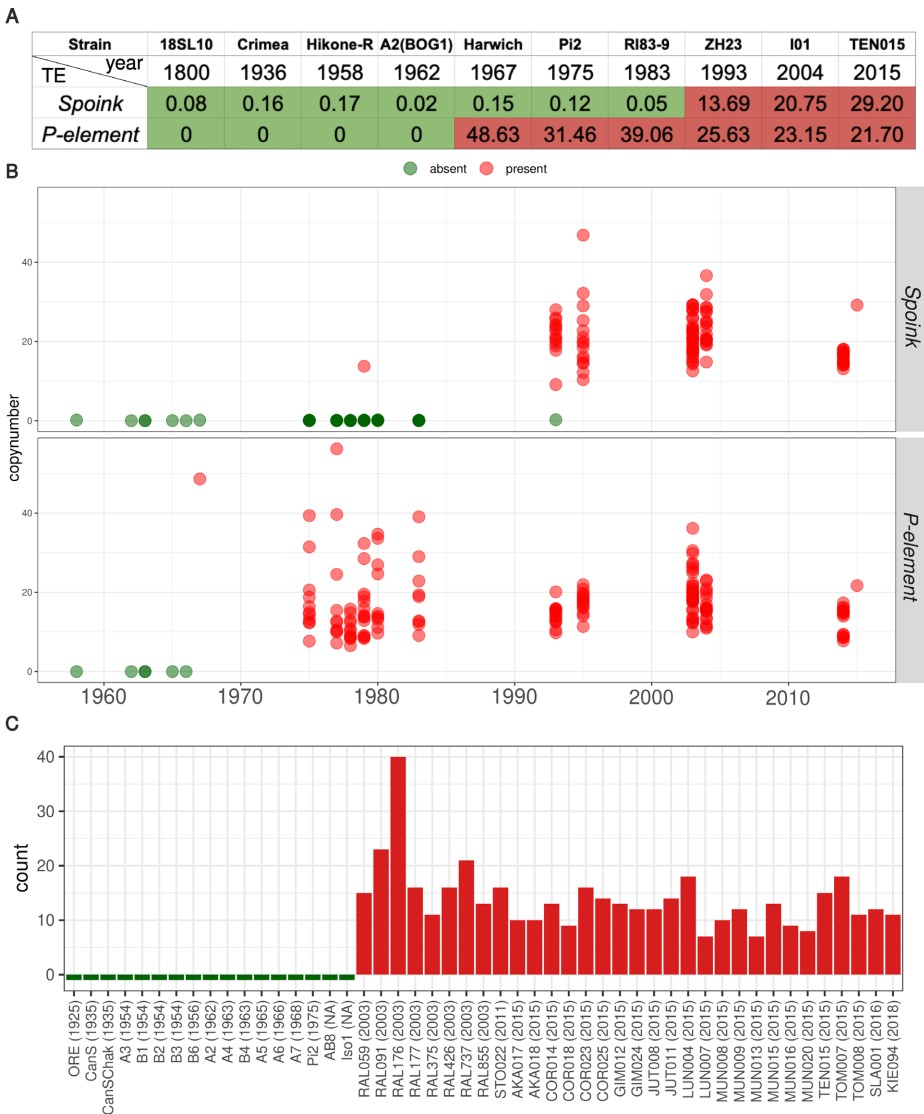

**Fig 3. *Spoink* invaded *D. melanogaster* between 1983 and 1993 after the invasion of the *P*-element.** A) Rough timeline of the *Spoink* and *P*-element invasion based on different strains sampled during the last two hundred years. The numbers represent the estimated copy number of *Spoink* and *P*-element based on DeviaTE. B) Timeline of the *Spoink* and *P*-element invasion based on 183 strains sampled between 1960 and 2015. The intensity of the color varies due to overlapping dots C) Abundance of canonical *Spoink* insertions (> 80% length and < 5% divergence) in long-read assemblies of *D. melanogaster* strains collected between 1925 and 2018.

these strains using DeviaTE [55]. As reference we also estimated the abundance of the *P*-element, which is widely assumed as to be the most recent TE that invaded *D. melanogaster* populations [28, 31]. *Spoink* was absent from all strains collected ≤1983 but present in strains collected ≥1993 (Fig 3A). By contrast our data suggest that the *P*-element was absent in the strains collected ≤ 1962 but present in strains collected ≥1967 (Fig 3A). This is consistent with previous works suggesting that the *P*-element invaded *D. melanogaster* between 1950 and 1980 [21, 29, 35, 36]. Our data thus suggest that *Spoink* invaded *D. melanogaster* after the *P*-element invasion. To investigate the timing of the invasion in more detail we estimated the abundance of *Spoink* in short-read data of 183 strains collected between 1960 and 2015 from

different geographic regions using DeviaTE (S5 Table; data from [31, 40, 51–53]). The analysis of these 183 strains supports the view that *Spoink* was largely absent in strains collected ≤ 1983 but present in strains collected ≥ 1993 (Fig 3B). However there are two outliers. *Spoink* is present in one strain collected in 1979 in Providence (USA), which could be due to a contamination of the strain. On the other hand *Spoink* is absent in one strain collected in 1993 in Zimbabwe (Fig 3B). As *Spoink* was present in six other strains collected in 1993 from Zimbabwe, it is feasible that *Spoink* was still spreading in populations from Zimbabwe around 1993. The strains supporting the absence of *Spoink* prior to 1983 were collected from Europe, America, Asia and Africa while the strains supporting the presence of *Spoink* after 1993 were collected from all five continents (S5 Table).

Finally we estimated the abundance of *Spoink* in 49 long-read assemblies of strains collected during the last 100 years (S6 Table; [39, 40, 48, 56]). We used RepeatMasker [37] to estimate the abundance of canonical *Spoink* insertions (> 80% length and < 5% divergence) in these strains. Canonical *Spoink* insertions were absent in strains collected before 1975 but present in all long-read assemblies of strains collected after 2003 (Fig 3C). The strains of the assemblies supporting the absence of canonical *Spoink* insertions were collected from America, Europe, Asia, and Africa whereas the strains showing the presence of *Spoink* were largely collected from Europe, though genomes from North America and Africa are also represented (S6 Table).

In summary we conclude that *Spoink* invaded worldwide populations of *D. melanogaster* approximately between 1983 and 1993. Moreover, the *Spoink* invasion is more recent than the *P*-element invasion.

## Geographic heterogeneity in the *Spoink* sequence variation

Previous work showed that the composition of TEs within a species may differ among geographic regions [21, 31]. Such geographic heterogeneity could result from founder effects occurring during the geographic spread of a TE. For example, a TE spreading in a species with a cosmopolitan distribution such as *D. melanogaster* may need to overcome geographic obstacles such as oceans and deserts. The few individuals that overcome these obstacles, thereby spreading the TE into hitherto naive populations, may carry slightly different variants of the TE than the source populations. These distinct variants will then spread in the new population. Such founder effects during the invasion may lead to a geographically heterogeneous composition of a TE within a species. For example, for the retrotransposon *Tirant*, individuals sampled from Tasmania carry distinct variants [31], while for *412* and *Opus* individuals from Zimbabwe are distinct from the other populations [21]. To investigate whether we find such geographic heterogeneity we analysed the *Spoink* composition in the Global Diversity Lines (GDL), which comprise 85 *D. melanogaster* strains sampled after 1988 from five different continents (Africa—Zimbabwe, Asia—Beijing, Australia—Tasmania, Europe—Netherlands, America—Ithaca; [51]). Except for a single strain from Zimbabwe all GDL strains harbour *Spoink* insertions (S5 Fig). We estimated the allele frequency of SNPs in *Spoink*, where a SNP refers to a variant among dispersed copies of *Spoink*. The allele frequency estimate thus reflects the composition of *Spoink* within a particular strain. To summarize differences in the composition among the GDL strains we used UMAP [76]. We found that the composition of *Spoink* varies among regions where three distinct groups can be distinguished: Tasmania, Beijing/Ithaca and Netherlands/Zimbabwe (S5 Fig). It is interesting that clusters are formed by geographically distant populations such as Beijing (Asia) and Ithaca (America). We speculate that human activity, where flies might for example hitchhike with merchandise, could be responsible for this pattern. In summary, we found a geographically heterogeneous composition of *Spoink* which is likely due to founder effects occurring during the spread of this TE.

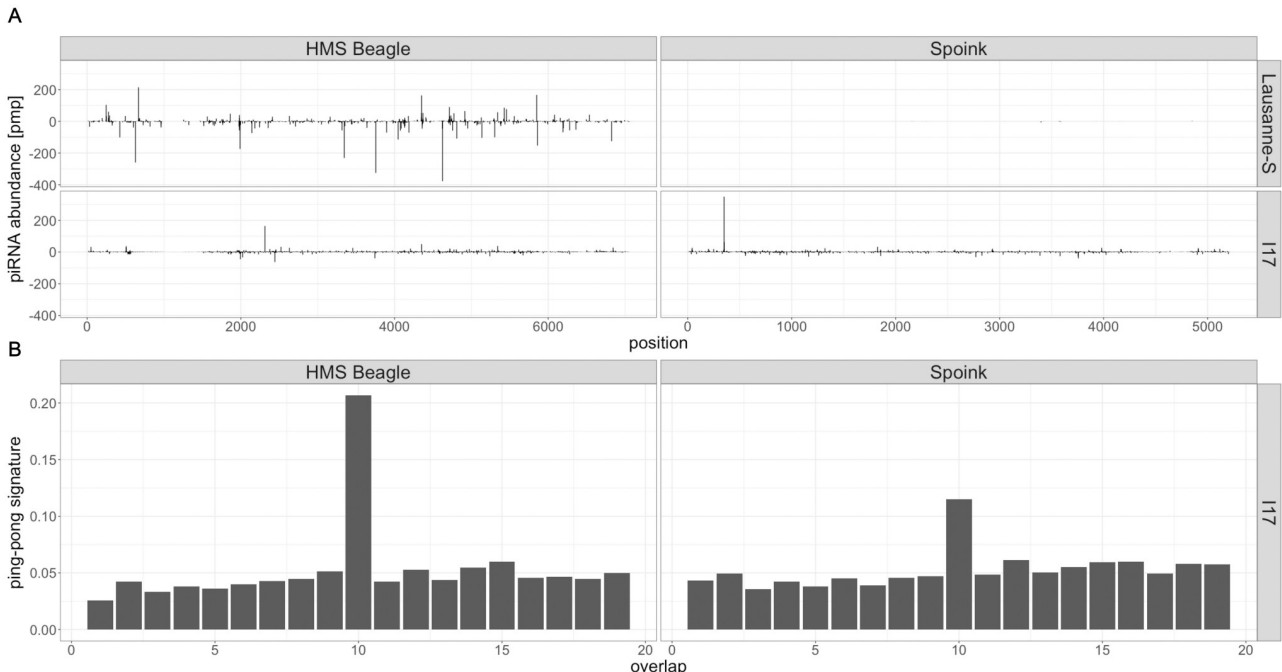

**Fig 4.** A piRNA based defence against *Spoink* emerged in *D. melanogaster* A) piRNAs mapping to *Spoink* in a strain sampled 1938 (*Lausanne-S*) and 2004 (*I17*). The transposon *HMS Beagle* is included as reference. Solely the 5' positions of piRNAs are shown and the piRNA abundance is normalized to one million piRNAs. Sense piRNAs are shown on the positive y-axis and antisense piRNAs on the negative y-axis. B) Ping-pong signature for the piRNAs mapping to *Spoink* and *HMS Beagle* in the *D. melanogaster* strain *I17* (2004).

## *Spoink* is silenced by the piRNA pathway in natural populations

The host defence against TEs in *Drosophila* is based on small RNAs termed piRNAs. These piRNAs bind to PIWI clade proteins and silence a TE at the transcriptional as well as the post-transcriptional level [11, 12, 14, 77]. To test whether *Spoink* is silenced in *D. melanogaster* populations we investigated small RNA data from the GDL lines [62]. Small RNA were sequenced for 10 out of the 84 GDL lines such that two strains were picked from each of the five continents [62].

We find piRNAs mapping along the sequence of *Spoink* in the GDL strain *I17* which was collected in 2004 but not in the strain *Lausanne-S* which was sampled around 1938 (Fig 4A; [78]). piRNAs mapping to *Spoink* were further found for all 10 GDL strains (S6 Fig).

An important feature of germline piRNA activity in *D. melanogaster* is the ping-pong cycle [11, 12]. An active ping-pong cycle generates a characteristic overlap between the 5' positions of sense and antisense piRNAs, i.e. the ping-pong signature. Computing a ping-pong signature thus requires several overlapping sense and antisense piRNAs. Since the amount of piRNAs was too low we could not compute a ping-pong signature for the strain *Lausanne-S* (collected in 1938; see above). However we found a pronounced ping-pong signature in all 10 GDL samples (Fig 4B and S6 Fig).

It is an important open question as to which events trigger the emergence of piRNA based host defence. The prevailing view, the trap model, holds that the piRNA based host defence is initiated by a copy of the TE jumping into a piRNA cluster [17, 25, 79–81]. If this is true we expect *Spoink* insertions in piRNA clusters in each of the long-read assemblies of the recently collected *D. melanogaster* strains [40]. We identified the position of piRNA clusters in these

long-read assemblies based on unique sequences flanking the piRNA clusters [48]. Interestingly, we found an extremely heterogeneous abundance of *Spoink* insertions in piRNA clusters, where some strains (e.g. *RAL176*) have up to 14 cluster insertions whereas 18 out of 31 strains did not have a single cluster insertion (S7 Table). Three of the cluster insertions were into *42AB*, which usually generates the most piRNAs [11, 69]. It is an important open question whether such a heterogeneous distribution of *Spoink* insertions in piRNA clusters is compatible with the trap model [82, 83]. In summary we found evidence that *Spoink* is silenced by the piRNA pathway but the number of *Spoink* insertions in piRNA clusters is very heterogeneous among strains.

## Origin of *Spoink*

The invasion of *Spoink* in *D. melanogaster* was likely triggered by horizontal transfer from a different species. To identify the source of the horizontal transfer we investigated the long-read assemblies of 101 *Drosophila* species [67] (and *D. simulans* strain *SZ129*) and of 99 insect species [23, 67, 68] (S8 Table). We did not consider short-read assemblies, as TEs may be incompletely represented in them [48]. Apart from *D. melanogaster* we found insertions with a high similarity to *Spoink* in *D. sechellia*, in one out of two *D. simulans* assemblies (in SZ129 but not in 006), and species of the *willistoni* group, in particular *D. willistoni* (Fig 5A). In agreement with this, a sequence from *D. willistoni* with a high similarity to *Spoink* can be found in RepBase (*Gypsy-78_DWil*; I: 99.73% similarity, LTR: 93.54% similarity [84]). *Spoink* insertions with a somewhat smaller similarity were found in *D. cardini* and *D. repleta*. No sequences similar to *Spoink* were found in the 99 insect species (S7 Fig). To further shed light on the origin of the *Spoink* invasion we constructed a phylogenetic tree with full-length insertions of *Spoink* in *D. melanogaster*, *D. sechellia*, *D. simulans* (SZ129) *D. cardini* and species of the *willistoni* group (Fig 5B and for a star phylogeny see S8 Fig). We did not find a full-length insertion of *Spoink* in *D. repleta*. This tree reveals that *Spoink* insertions in *D. sechellia* and *D. simulans* have very short branches. Furthermore, in *D. simulans* just one out of the two analysed assemblies has *Spoink* insertions. We thus suggest that the *Spoink* invasion in these two species is also of recent origin (manuscript in preparation).

However, *Spoink* insertions in *D. melanogaster* are nested within insertions from species of the *willistoni* group (Fig 5B). Our data thus suggest that, similar to the *P*-element invasion in *D. melanogaster* [27], the *Spoink* invasion in *D. melanogaster* was also triggered by horizontal transfer from a species of the *willistoni* group. The synonymous divergence of *Spoink* is lower than for any of 140 single copy orthologous genes shared between *D. melanogaster* and *D. willistoni*, further supporting the recent horizontal transfer of *Spoink* (S9 Fig) [20, 85, 86]. Species of the *willistoni* group are Neotropical, occurring throughout Central and South America [87–89]. Therefore horizontal transfer of *Spoink* only became feasible after *D. melanogaster* extended its habitat into the Americas approximately 200 years ago [90–92]. Insertions of *D. cardini* are next to species of the *willistoni* group, suggesting that *D. cardini* also acquired *Spoink* by horizontal transfer from the *willistoni* group, likely independent of *D. melanogaster* (Fig 5B). *D. cardini* is also a Neotropical species and its range overlaps many species of the *willistoni* group, thus horizontal transfer between the species is physically feasible [93, 94].

In summary, similarly to the *P*-element, horizontal transfer from a species of the *willistoni* group likely triggered the *Spoink* invasion in *D. melanogaster*.

## Discussion

Here we suggest that the LTR-retrotransposon *Spoink* invaded *D. melanogaster* populations between 1983 and 1993, after the spread of the *P*-element. Similarly to the *P*-element, the

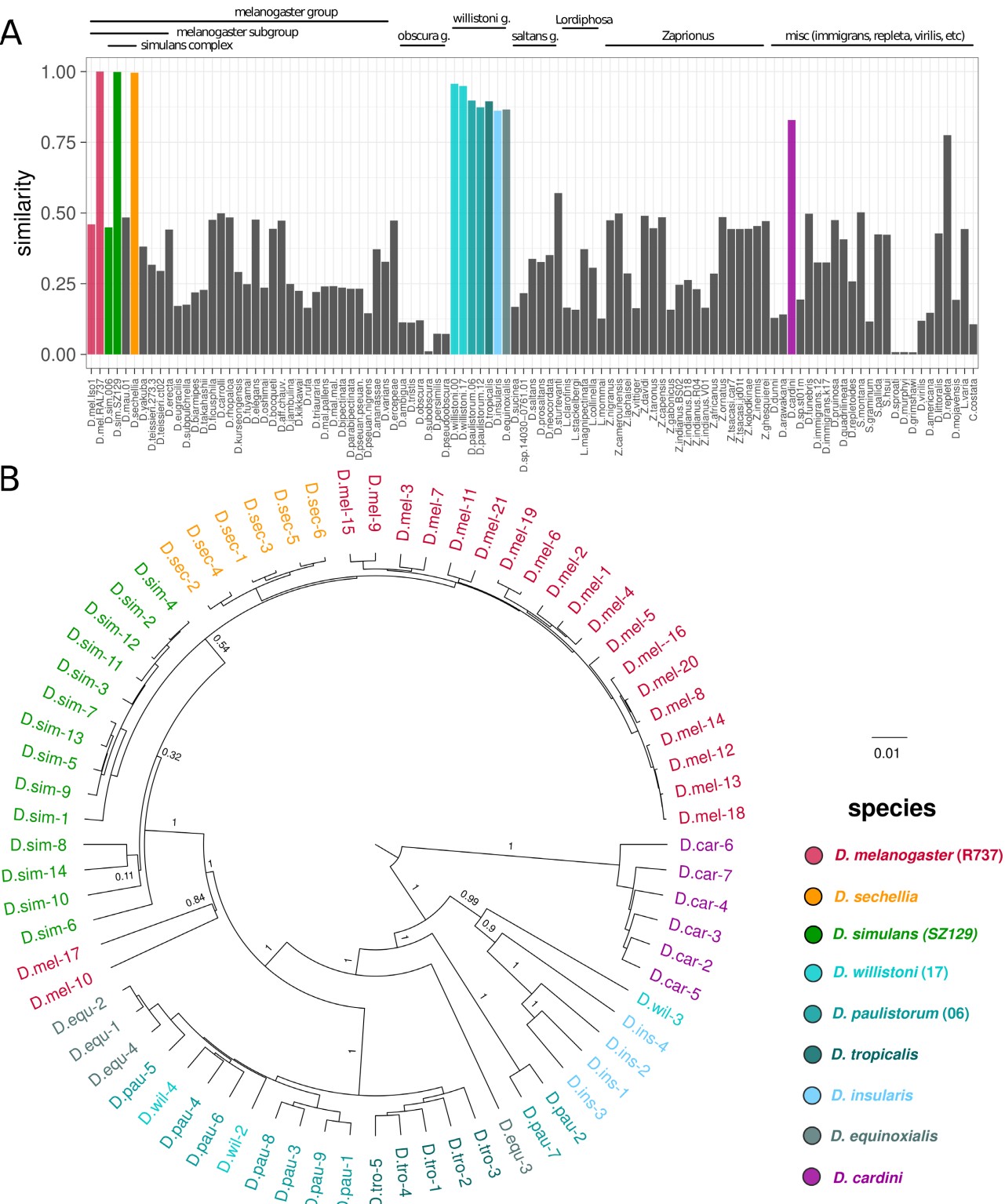

**Fig 5. The *Spoink* invasion in *D. melanogaster* was likely triggered by a horizontal transfer from a species of the *willistoni* group.** A) Similarity of TE insertions in long-read assemblies of diverse drosophilid species to *Spoink*. The barplots show for each species the similarity between *Spoink* and the best match in the assembly. For example, a value of 0.9 indicates that at least one insertion in an assembly has a high similarity (≈ 90%) to the consensus sequence of *Spoink*. B) Bayesian tree of *Spoink* insertions in the different drosophilid species. Only full-length insertions of *Spoink* (> 80% of the length) were considered. Node support values are posterior probabilities estimated by BEAST [45]. Note that *Spoink* insertions of *D. melanogaster* are nested in insertions from the *willistoni* group (blue shades).

*Spoink* invasion was likely triggered by horizontal transfer from a species in the *willistoni* group. Horizontal transfer of a TE is usually inferred from three lines of evidence: i) a patchy distribution of the TE among closely related species, ii) a phylogenetic discrepancy between the TE and the host species and iii) a high similarity between the TE of the donor and recipient species, which is frequently quantified by the synonymous divergence of the TE [95, 96]. All of these three lines of arguments support a horizontal transfer of *Spoink* in *D. melanogaster*, with a species of the *willistoni* group being the likely donor. First we found a patchy distribution among species of the *melanogaster* group (for *D. simulans* we even have a patchy distribution among different strains; Fig 5A). Second *Spoink* insertions of *D. melanogaster* (and other species that may have gotten *Spoink* recently) are nested within species of the *willistoni* group (Fig 5B), a clear phylogenetic discrepancy. Third we found that the synonymous divergence of *Spoink* is lower than for all orthologous genes in *D. melanogaster* and *D. willistoni* (S9 Fig). In addition to this classical but indirect lines of evidence, we have however more direct and thus more compelling evidence for the horizontal transfer of *Spoink*. Based on strains collected during the last hundred years from all major geographic regions we showed that *Spoink* insertions were absent in all strains collected before 1983 but present in all strains collected after 1993 (using Illumina short read data, long-read assemblies, and PCR/Sanger sequencing). This makes *Spoink* one of the best documented cases of a recent horizontal transfer of a TE, similarly to the *P*-element where also strains collected during the last 100 years support the recent horizontal transfer [28, 29].

The abundance of sequencing data from strains collected at different time points during the last century allowed us to pinpoint the timing of the invasion in a way that would not have been previously possible. *Spoink* appears to have rapidly spread throughout global populations of *D. melanogaster* between 1983 and 1993. The narrow time-window of 10 years is plausible as studies monitoring *P*-element invasions in experimental populations showed that the *P*-element can invade populations within 20–60 generations [65, 97, 98]. Assuming that natural *D. melanogaster* populations have about 15 generations per year [99], a TE could penetrate a natural *D. melanogaster* population within 1–3 years. Given this potential rapidness of TE invasions it is likely that *Spoink* spread quickly between 1983 and 1993. Since there is a gap between strains sampled at 1983 and 1993 we cannot further narrow down the timing of the invasion. Furthermore, the strains used for timing the invasions were sampled from diverse geographic regions and *Spoink* likely spread at different times in different geographic regions. If horizontal transfer from a *willistoni* species triggered the invasion, as suggested by our data, then *Spoink* will have first spread in *D. melanogaster* populations from South America (the habitat of *willistoni* species), followed by populations from North America and the other continents. It is also feasible that *Spoink* invaded *D. melanogaster* indirectly, for example using *D. simulans* as intermediate host, in which case the *Spoink* invasion in *D. melanogaster* may have been triggered in almost any geographic region (both, *D. simulans* and *D. melanogaster*, are cosmopolitan species [100]). Unfortunately, we cannot infer the timing of the geographic spread of the *Spoink* invasion in different continents as *D. melanogaster* strains were not sampled sufficiently densely from different regions. Our work thus highlights the importance of efforts such as DrosEU, GDL and DrosRTEC to densely sample *Drosophila* strains in time and space [51, 101, 102]. It is also interesting to ask as to which extent human activity (e.g. trafficking of goods) contributed to the rapid spread of *Spoink*. Given that our analysis of the *Spoink* composition shows that geographically distant populations (Bejing/Ithaca or Netherlands/Zimbabwe) cluster together, human activity may have played a role. Increasing human activity could also explain why *Spoink* (invasion 1983–1993) seems to have spread faster than the *P*-element (1950–1980).

Our investigation of *Spoink* insertions in different drosophilid species suggests that the *Spoink* invasion in *D. melanogaster* was triggered by horizontal transfer from a species of the *willistoni* group. Although it is possible that we did not analyse the true donor species, we consider it unlikely to be a species outside of the *willistoni* group given the wide distribution of *Spoink* in all species in the *willistoni* group. In addition, the phylogenetic tree of *Spoink* has deep branches within the *willistoni* group, suggesting that *Spoink* is ancestral in this group (S10 Fig).

A related open question is when *Spoink* first entered *D. melanogaster* populations. Since a TE may initially solely spread in some isolated subpopulations there could be a considerable lag time between the horizontal transfer of a TE and its spread in worldwide population. The presence of *Spoink* in a strain collected around 1979 in Providence (USA; Fig 3B) could be due to this lag time (or contamination). Nevertheless, the horizontal transfer of *Spoink* must have happened between the spread of *D. melanogaster* into the habitat of the *willistoni* group, about 200 years ago, and the invasion of *Spoink* in worldwide populations between 1983 and 1993. In addition to the *P*-element, *Spoink* is the second TE that invaded *D. melanogaster* populations following horizontal transfer from a species of the *willistoni* group. Species from the *willistoni* group are very distantly related with *D. melanogaster* (about 100my [103]) and we were thus wondering whether it is a coincidence that a species of the *willistoni* group is again acting as donor of a TE invasion in *D. melanogaster*. The recent habitat expansion of *D. melanogaster* into the Americas resulted in novel contacts with many species, in addition to species of the *willistoni* group, that might have acted as donors of novel TEs such as *D. pseudoobscura* or *D. persimilis* [104]. Why is again a species of the *willistoni* group and not one of these other species acting as donor of a novel TE? Apart from mere chance, there are several, not mutually exclusive, hypotheses for this observation. First, it is feasible TEs of the *willistoni* group are exceptionally compatible with *D. melanogaster* at a molecular level. Second, some parasites targeting both *D. melanogaster* and species of the *willistoni* group could be efficient vectors for horizontal transfer of TEs. Third, the physical contact between *D. melanogaster* and some species of the *willistoni* group might be unusually tight, facilitating horizontal transfer of TEs by an unknown vector. *D. willistoni* is a common drosophilid in South American forests [105]. Habitat fragmentation caused by human deforestation may thus generate intensive contacts between human commensal species, such as *D. melanogaster*, and abundant forest species like *D. willistoni*. Fourth, species of the *willistoni* group might be exceptionally numerous resulting in elevated probability for horizontal transfer of a TE.

The *Spoink* invasion is the eighth TE invasion in *D. melanogaster* that has occurred during the last 200 year. As we argued previously, such a high rate of TE invasions is likely unusual during the evolution of the *D. melanogaster* lineage since the number of TE families in *D. melanogaster* is much smaller than what would be expected if this rate of invasions would persist [21]. It is possible that the high rate of TE invasions continues beyond the past 200 years since many LTR transposons in *D. melanogaster* are likely of very recent origin (possibly $< 16.000$ years [85, 106]). One possible explanation for this high rate of recent TE invasions is that human activity contributed to the habitat expansion of *D. melanogaster*. Due to this habitat expansion *D. melanogaster* spread into the habitat of *D. willistoni* which enabled the horizontal transfer of *Spoink*. This raises the possibility that other species with recent habitat expansions also experienced unusually high rates of TE invasions. It is also interesting to ask whether the rate of TE invasions differs among species. For example cosmopolitan species, such as *D. melanogaster*, may generally experience higher rates of horizontal transfer than more locally confined species. The cosmopolitan distribution will bring species into contact with many diverse species, thereby increasing the opportunities for horizontal transfer of a TE.

The *Spoink* invasions also opens up several novel opportunities for research. First, the broad availability of strains with and without *Spoink* will enable testing whether *Spoink* activity induces phenotypic effects, similarly to hybrid dysgenesis described for the *P*-element, *I*-element and *hobo*, but not for *Tirant* [31, 107–109]. Second, it will be interesting to investigate whether some *Spoink* insertions participated in rapid adaptation of *D. melanogaster* populations, similar to a *P*-element insertion which contribute to insecticide resistance [110]. Third, it will enable studying *Spoink* invasions in experimental populations, shedding light on the dynamics of TE invasions, much as other recent studies investigating the invasion dynamics of the *P*-element [97, 98, 111]. Fourth, investigation into the distribution of species that have been infected with *Spoink* will shed light on the networks of horizontal transfer in drosophilid species. Fifth, the *Spoink* invasion provides an opportunity to study the establishment of the piRNA-based host defence [similar to [24, 65]]. For example we found that none of the piRNA cluster insertions are shared between individuals, suggesting there is no or solely weak selection for piRNA cluster insertions. Furthermore we found an extremely heterogeneous abundance of *Spoink* insertions in piRNA clusters where we could not find a single cluster insertions of *Spoink* in several strains. It is an important open question whether such a heterogeneous distribution is compatible with the trap model [83]. One possibility is that a few cluster insertions in populations are sufficient to trigger the paramutation of regular (non-paramutated) *Spoink* insertions into piRNA producing loci [16, 112, 113]. These paramutated *Spoink* insertions may then compensate for the low number of *Spoink* insertions in piRNA-clusters [112]. Paramutations could thus explain why several studies found that stand-alone insertions of TEs can nucleate their own piRNA production [69, 83, 114, 115].

The war between transposons and their hosts is constantly raging, with potentially large fitness effects for the individuals in populations. Over the last two hundred years there have been at least eight invasions of TEs into *D. melanogaster*, each of which could disrupt fertility for example by inducing some form of hybrid dysgenesis. TEs are responsible for > 80% of visible spontaneous mutations in *D. melanogaster*, and produce more variation than all SNPs combined [116–118]. In the long read assemblies considered here, more than half of insertions of *Spoink* were into genes [40]. The recent *Spoink* invasion could thus have a significant impact on the evolution of *D. melanogaster* lineage.

## Supporting information

**S1 Fig. Abundance of *Spoink* and *P*-element insertions in different genomic features.** TE insertions were identified in 31 long-read assemblies of *D. melanogaster* [40] and the reference annotation was lifted to each assembly with liftoff [46, 47]. Note that the P-element has a pronounced insertion bias in promoters (defined as 1000bp upstream of the first exon) whereas *Spoink* insertions are largely found in introns and intergenic regions.
(AI)

**S2 Fig. DeviaTE plots of six *D. melanogaster* strains collected during the last century.** The short reads were aligned to the consensus sequence of *Spoink* and the coverage was normalized to the the coverage of single-copy genes. The coverage was manually curbed at the poly-A track (indicated by dashed lines). Note that very few reads of old strains ($\leq$ 1975) align to *Spoink* whereas a contiguous coverage of reads along *Spoink* is observed for more recently collected strains ($\geq$ 1993).
(AI)

**S3 Fig. Abundance of *Spoink* insertions in six long-read assemblies of *D. melanogaster* strains collected during the last century.** Note that all strains contain fragmented and

diverged insertions of *Spoink*, while solely recently collected strains (≥2003) contain canonical *Spoink* insertions (i.e. full-length insertions with little divergence from the consensus sequence).
(AI)

**S4 Fig. The Sanger sequence of the six PCR amplicons matches with the consensus sequence of *Spoink*.** The Sanger sequences of the amplicons of P1 (red) and P2 (green) have been aligned to the consensus sequence of *Spoink* (blue, top) and the coordinates of the alignments are indicated. The *D. melanogaster* strain and the sequence similarity between the Sanger sequence and the consensus sequence of *Spoink* are provided next to each matching region.
(SVG)

**S5 Fig. Abundance and composition of *Spoink* insertions in the GDL.** A) Abundance of *Spoink* in the GDL. Note that one strain from Zimbabwe does not have any *Spoink* insertion. B) UMAP summarizing the composition of *Spoink* among the GDL. Note that *Spoink* shows a pronounced population structure, where three main clusters can be discerned: Tasmania, Bejing/Ithaca and Netherlands/Zimbabwe.
(SVG)

**S6 Fig. A piRNA based defence against *Spoink* is active in the 10 GDL strains.** Two strains are analysed for each continent (Bxx Beijing/Asia, Ixx Ithaca/America, Nxx Netherlands/Europe, Txx Tasmania/Australia, ZWxx Zimbabwe/Africa; the second strain from Ithaca (*I17*) is in the main manuscript). A) piRNAs mapping to the sequence of *Spoink*. Solely the 5' positions of piRNAs are shown and the piRNA abundance is normalized to one million piRNAs. Sense piRNAs are shown on the positive y-axis and antisense piRNAs on the negative y-axis. B) ping-pong signature of *Spoink*.
(SVG)

**S7 Fig. Barplots show the similarity between the consensus sequence of *Spoink* and the best match in each of 99 long-read assemblies of diverse insect species.** As a reference, two *D. melanogaster* assemblies (red) were included, where D.mel.RAL176 has canonical *Spoink* insertions while D.mel.Iso1 solely has degraded fragments of sequences having some similarity with *Spoink*.
(AI)

**S8 Fig. Star phylogeny of *Spoink* insertions in the different drosophilid species.** Only full-length insertions of *Spoink* (> 80% of the length) were considered.
(SVG)

**S9 Fig. Distribution of synonymous divergence for *Spoink* and 140 single copy orthologous genes shared between *D. melanogaster* and *D. willistoni* (red).** For *Spoink* we used the shared part of the longest ORF (green). The red dashed line is the 2.5% quantile of nuclear genes [85]. Note that the dS of *Spoink* is lower than the dS of any of the orthologous genes shared between *D. melanogaster* and *D. willistoni*, consistent with a horizontal transfer of *Spoink* between the two species. The genes were obtained with the software BUSCO [119]. The predicted proteins were aligned using Clustal Omega [120]. The codons information from the protein alignment was used for the nucleotide alignment using PAL2NAL [121]. The dS was calculated using the software PAML.
(PNG)

**S10 Fig. Average distance between 100 pairs of *Spoink* insertions randomly sampled within either the *melanogaster* group (i.e *D. melanogaster*, *D. simulans*, *D. sechellia*) or the *willistoni* group.** Distances within the *willistoni* group are significantly longer than the distances in the *melanogaster* group ($t = −6.31$, $df = 193.88$, $p = 1.762e − 09$). Note that this test accounts for the phylogenetic information of the tree using the distances of the insertions within the two groups.
(PNG)

**S1 Table. Differences in the abundance of *Gypsy_7_DEl* between the reference genome Iso1 and a long-read assemblies from a more recently collected strain.** The best ten matches for *Gypsy_7_DEl* and the consensus sequence of *Spoink* are shown for both assemblies. Matches were identified with RepeatMasker [37]. Note that the discrepancy between Iso1 and TOM007 is more pronounced when the consensus sequence of *Spoink* is considered.
(XLSX)

**S2 Table. Similarity between *Spoink* and other TEs in the different repeat libraries generated for *D. melanogaster*.** For each repeat library the best five hits are shown. Solely matches with a minimum overlap with *Spoink* of at least 30% are considered. subst. substitions in percent between *Spoink* and the TE, len. fraction of the length of a TE aligning with *Spoink*; a [40], b [43], c [5], d [69], e [70].
(XLSX)

**S3 Table. Coordinates of *Spoink* insertions in the strains RAL091, RAL176 and RAL732 used for Fig 1A of the main manuscript.**
(XLSX)

**S4 Table. Identity of sequences in Oregon-R having some similarity with the consensus sequence of *Spoink*.** Solely sequences having a divergence of ≤25% and minimum overlap of at least 10% with *Spoink* are considered. The sequences were extracted from the assembly of Oregon-R (chromosome:start-end) and aligned against the TE library of *D. melanogaster* using blastn [43, 122]. Most of these sequences match TARTC and DMDM11.
(XLSX)

**S5 Table. Overview of the short-read data analysed in this work.** Data are from [31, 40, 51–53]).
(XLSX)

**S6 Table. Overview of the long-read assemblies of *D. melanogaster* strains analysed in this work.** For each strain we show the assembly ID, the strain, the sampling location and the sampling date. a [38, 39], b [48], c [56], d [40], e [123].
(XLSX)

**S7 Table. *Spoink* insertions in piRNA clusters of long-read assemblies of different *D. melanogaster* strains [40].** Note that for several strains we could not find a single *Spoink* insertion in a piRNA cluster. On the other hand, some strains, like RAL176, have multiple *Spoink* insertions in piRNA clusters.
(XLSX)

**S8 Table. Overview of the long-read assemblies of diverse insect species analysed in this work.**
(XLSX)

## Acknowledgments

We thank Matthew Beaumont for the idea to call the here described transposon *Spoink*. We thank Silke Jensen for comments. We thank Neda Barghi and Claudia Ramirez Lanzas for providing fly strains used for PCR. SS would like to thank J. B. Signor for helpful comments on the manuscript. RK, RP, and AS thank all members of the Institute of Population Genetics for feedback and support.

## Author Contributions

**Conceptualization:** Sarah Signor, Robert Kofler.

**Data curation:** Riccardo Pianezza, Almorò Scarpa, Robert Kofler.

**Formal analysis:** Riccardo Pianezza, Almorò Scarpa, Sarah Signor, Robert Kofler.

**Funding acquisition:** Sarah Signor, Robert Kofler.

**Investigation:** Almorò Scarpa, Prakash Narayanan, Sarah Signor, Robert Kofler.

**Methodology:** Riccardo Pianezza.

**Project administration:** Sarah Signor, Robert Kofler.

**Resources:** Sarah Signor, Robert Kofler.

**Software:** Riccardo Pianezza, Robert Kofler.

**Supervision:** Sarah Signor, Robert Kofler.

**Visualization:** Riccardo Pianezza, Almorò Scarpa, Sarah Signor, Robert Kofler.

**Writing – original draft:** Riccardo Pianezza, Almorò Scarpa, Sarah Signor, Robert Kofler.

**Writing – review & editing:** Riccardo Pianezza, Almorò Scarpa, Sarah Signor, Robert Kofler.

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
