## [Decision Letter · Decision Letter 0]

1 Feb 2024

Dear Dr. Koefler,

Thank you very much for submitting your Research Article entitled 'Spoink, a LTR retrotransposon, invaded D. melanogaster populations in the 1990s' to PLOS Genetics.

The manuscript was fully evaluated at the editorial level and by three independent peer reviewers. The reviewers appreciated the attention to an important topic but identified some concerns that we ask you address in a revised manuscript. In particular I want to draw your attention to the comments of Reviewers 2 and 3 regarding the presence of closely related elements in other Drosophila species (including some apparently not reported in your manuscript) which should be carefully considered in your inference/discussion of the donor species and the evolutionary history of this element in general. These are compulsory points that should be addressed in your revision. 

We therefore ask you to modify the manuscript according to the review recommendations. Your revisions should address the specific points made by each reviewer.

Yours sincerely,

Cédric Feschotte

Academic Editor

PLOS Genetics

Justin Fay

Section Editor

PLOS Genetics

Reviewer's Responses to Questions

**Comments to the Authors:**

Reviewer #1: This study reports a new gypsy-like retrotransposon which has apparently invaded populations of Drosophila melanogaster through recent horizontal transfer taking place between 1983 and 1993. The source of the transfer may be a Drosophila species belonging to the willistoni group, which would place the transfer somewhere in South America after D. melanogaster reached this continent. I enjoyed reading the paper. It is well-written, with well-crafted, informative figures. I agree with the authors that this is an important and interesting observation that opens many new questions and that will be useful to further study the causes and consequences of horizontal transfer through multiple angles. I am thus overall enthusiastic and would like to congratulate the authors for this work. I would however also like them to take into considerations a few comments.

1 – I am wondering why the search for Spoink in other Drosophila species was limited to long-read assemblies. When performing a blastn search using one Spoink insertion (CM034735.1 6915682 6920905) as query on all Drosophila assemblies available in Genbank I found many insertions >90% identical to this insertion in species not included in fig 5 (gaucha, nebulosa, fulvimacula, leonis, medidiana, meridionalis, anceps) as well as in species in which Spoink is reported to be absent in fig 5 (e.g. in multiple assemblies of D. simulans). Some of these assemblies may have been deposited in Genbank after the study was conducted, but not all. The presence of multiple insertions >99% identical to Spoink in multiple simulans populations is puzzling and I would think it would be pertinent to have a closer look at them and include them in the study. Similarly, I found several copies of Spoink in D. paulistorum that show a higher ID (from 99.07 to 99.5%) to D. melanogaster Spoink than shown in fig 5. This might be due to the fact that I used a copy rather than the consensus but I would like to encourage the authors to further consider these paulistorum copies as they may help refine their scenario of transfer, perhaps suggesting that more than one transfer occurred, including some events more recent than 1993? The paulistorum copies I am referring to are: JAECXG010000750.1 67337 to 71947, JAECWZ010000441.1 146657 to 151263, JAECXG010000663.1 1412470 to 1417089.

2 – The presence of Spoink in one strain collected in 1979 in Providence (USA) is quickly dismissed as a possible contamination. Is there any other information that may help support this hypothesis? When was this Providence strain sequenced and where? Does it have the P element? Any genomic feature that may help support contamination? Has contamination between Drosophila strains been noticed before?

Minor comments listed following the order of the manuscript:

1 – Line 32, “termed piRNAs, which are”?

2 – Line 77: a few mismatches between which two sequences? Do you mean an imperfect poly-A tract?

3 – Line 79: providing evalues without mentioning which similarity search was done on which database is not very informative.

4 – Line 80: RNase H (not HI)?

5 – Line 81: env (lower case italics) may refer to the gene rather than the protein, not sure whether this comply with PLOS Genetics rules.

6 – Caption figure 1: use LTR retrotransposon rather than LTR transposon to fit with the core text?

7 – Line 89: above it is written that the gag pol is typical for ty3/gypsy. Is it typical or atypical and if atypical please elaborate?

8 – Line 156: collected in 1993

9 – Line 179: will then spread, not than

10 – figure 3: please add a color legend and perhaps mention that the intensity of color in the B panel varies because dots appear superimposed on each other? Also pleas eexplain what are the number in the table in panel A.

11 – Line 205: were further instead of we further

12 – Fig 5 b: I would personnally much refer to sea a star tree (rather than circular) with branch lengths proportional to inferred distance and a legend for branch lengths. Please consider adding simulans and other paulistorum insertions in this analysis as suggested in my earlier comment.

13 – Line 261: 15 generations

14 – Line 267: then Spoink will have

15 – Line 273: to which extent human activity

16 – Regarding the possible transfer from willistoni group to melanogaster, are there any ecological feature of melanogaster and willistoni sp that may support close physical contact between these species?

17 – Line 323: piRNA-based

18 – Line 331: may then compensate

19 – Line 429: please add a dot at the end of the sentence.

20 – Please consider submitting Spoink consensus sequence to repbase and other relevant databases.

Reviewer #2: Review of Pianezza et al “Spoink, a LTR retrotransposon, invaded D. melanogaster populations in the 1990s”

This manuscript provides a thorough investigation of a clear case of horizontal transfer of a previously uncharactereized LTR retrotransposon in one of the most well-studied model organisms, D. melanogaster. The evidence provided is rigorous and compelling and I only have relatively minor comments that could improve the work.

General comments

1) It would be helpful if the authors could review the required evidence to demonstrate horizontal transfer of a transposon, and then detail which pieces of evidence presented in the paper correspond to the requirements to demonstrate HTT. Wallau et al. 2012 GBE and Peccoud et al. 2018 Bioessays provide convenient reviews that summarize HTT evidence requirements. Interestingly, the authors also have an opportunity to formally extend the previously considered categories of evidence that support HTT by adding presence/absence across a time series as a new category of evidence (which the authors satisfy in the current work).

2) The authors should discuss the current findings in the context of prior genomic evidence for LTR retrotransposon insertions being young in in D. melanogaster by Bergman and Bensasson 2007 PNAS and comparative genomic evidence by Bartolome et al 2008 Genome Biology, which suggest that widespread HTT of LTR retrotransposons is common in D. melanogaster and has been an ongoing process for longer than the last 200 years.

3) The authors make the claim that it is surprising that HTT of Spoink went unnoticed because D. melanogaster is investigated heavily at the genetic level. However, this is not surprising to me because only a limited number of lab strains of D. melanogaster strains were intensively studied for the last 100 years, and complete genomes of natural populations have only been investigated very recently.

4) The authors often compare results from Spoink to the P-element. However, the P-element is arguable not the best control to contrast with distribution and variation in Spoink since it is a DNA transposon, not an LTR retrotransposon. I would urge the authors to add control analyses to the main or supplemental text comparing results for Spoink to another Ty3/gypsy LTR retrotransposon.

Specific comments

Line 32: change “are cognate” -to“that are cognate”

Line 79: change “e-value e=” to “e-value”

Lines 79, 80, 81: change “e =” to “e-value =”

Line 89: explain how gag-pol polyprotein for Spoink is atypical for Ty3/gypsy elements.

Line 107: change “of single copy” to “a sample of single copy”

Line 136: change “present in all” to “present in centric heterochromatic regions of all”

Line 155-6: Is there any other genome-wide evidence that the Providence 1979 strain is contaminated?

Line 172: change “Spoink composition” to “Spoink sequence variation”

Line 181-2: cite or provide evidence for Blood and Opus being different in Zimbabwe populations

Line 205: change “we further” to “were”

Line 224: change “we found that Spoink is silenced” to “we found evidence that Spoink is silenced”

Line 301: change “identified TE invasion in D. melanogaster during the last” to “TE invasion in D. melanogaster that has occurred during the last”

Line 351: change “Characterisation” to “Structure and Classification”

Line 385: which version of the TE database from Quesneville was used in the current analysis? Was this obtained from flybase or https://github.com/bergmanlab/drosophila-transposons?

Line 402-3 should “read” and “reads” be replaced with “alignment” and “alignments” since according to my understanding flanking sequences are being aligned not reads.

Figure 1: Increase font size of sequence and labels in Fig1A and node support values in Fig 1B. Detail how node support values in Fig 1B were calculated in legend. Provide contig IDs and coordinates of Spoink insertions in Fig 1A.

Figure 5: Detail how node support values in Fig 5B were calculated in legend.

References: Genus names and abbreviations are not capitalized in several references.

Reviewer #3: Review is attached.

**Have all data underlying the figures and results presented in the manuscript been provided?**

Reviewer #1: Yes

Reviewer #2: None

Reviewer #3: Yes

PLOS authors have the option to publish the peer review history of their article (what does this mean?). If published, this will include your full peer review and any attached files.

Reviewer #1: **Yes: **Clément Gilbert

Reviewer #2: No

Reviewer #3: No

---

## [Editor Report · Decision Letter 1]

27 Feb 2024

Dear Dr Kofler,

We are pleased to inform you that your manuscript entitled "Spoink, a LTR retrotransposon, invaded D. melanogaster populations in the 1990s" has been editorially accepted for publication in PLOS Genetics. Congratulations!

Yours sincerely,

Cédric Feschotte

Academic Editor

PLOS Genetics

Justin Fay

Section Editor

PLOS Genetics

Comments from the reviewers (if applicable):

**Data Deposition**

http://datadryad.org/submit?journalID=pgenetics&manu=PGENETICS-D-23-01323R1

**Press Queries**

---

## [Editor Report · Acceptance letter]

1 Mar 2024

PGENETICS-D-23-01323R1 

Spoink, a LTR retrotransposon, invaded D. melanogaster populations in the 1990s 

Dear Dr Kofler, 

We are pleased to inform you that your manuscript entitled "Spoink, a LTR retrotransposon, invaded D. melanogaster populations in the 1990s" has been formally accepted for publication in PLOS Genetics! Your manuscript is now with our production department and you will be notified of the publication date in due course.

With kind regards,

Lilla Horvath

PLOS Genetics

On behalf of:
